

# Exogenous glutathione maintains the postharvest quality of mango fruit by modulating the ascorbate-glutathione cycle

Yan Zhou[1], Jiameng Liu[2], Qiongyi Zhuo[1], Keying Zhang[1], Jielin Yan[1], Bingmei Tang[1], Xiaoyun Wei[1], Lijing Lin[2] and Kaidong Liu[1]

[1] Life Science and Technology School, Lingnan Normal University, Zhanjiang, China
[2] Hainan Key Laboratory of Storage & Processing of Fruits and Vegetables, Agricultural Products Processing Research Institute, Chinese Academy of Tropical Agricultural Sciences, Zhanjiang, China

## ABSTRACT

**Background:** Mango fruit is prone to decay after harvest and premature senescence, which significantly lowers its quality and commercial value.

**Methods:** The mango fruit (*Mangifera indica* L.cv. Guixiang) was treated with 0 (control), 2, 5, and 8 mM of reduced glutathione (GSH) after harvest. The fruit was stored at 25 ± 1 °C for 12 days to observe the changes in the antioxidant capacity and postharvest quality.

**Results:** Compared with the control, the 5 mM GSH treatment significantly decreased the weight loss by 44.0% and 24.4%, total soluble solids content by 25.1% and 4.5%, and soluble sugar content by 19.0% and 27.0%. Conversely, the 5 mM GSH treatment increased the firmness by 25.9% and 30.7% on days 4 and 8, respectively, and the titratable acidity content by 115.1% on day 8. Additionally, the 5 mM GSH treatment decreased the malondialdehyde and hydrogen peroxide contents and improved the antioxidant capacity of mango fruit by increasing the superoxide dismutase and peroxidase activities and upregulating the expression of the encoding genes. Meanwhile, the higher levels of monodehydroascorbate reductase, dehydroascorbate reductase, and glutathione reductase enzyme activities and gene expressions accelerated the AsA-GSH cycle, thereby increasing the accumulation of AsA and GSH and maintaining the redox balance.

**Conclusions:** Overall, the experimental results suggest that 5 mM GSH maintains high antioxidant capacity and postharvest quality of mangoes and can use as an effective preservation technique for postharvest mangoes.

Corresponding authors
Yan Zhou, 286138826@qq.com
Kaidong Liu, liukaidong2001@126.com

## INTRODUCTION

As an economically important and globally cultivated fruit, mango (*Mangifera indica* L.) is consumed worldwide and is known for its flavor, shape, nutrition, delicious taste, and texture (*Rastegar, Hassanzadeh Khankahdani & Rahimzadeh, 2019*). Mangoes contain various nutrients, including minerals, vitamins, carotenoids, flavonoids, and fibers (*Wang et al., 2020*), with a high commercial value and climacteric. However, they have a limited

shelf life due to moisture loss, tissue softening, and fungal degeneration (*Singh et al., 2013*). Generally, harvested mangoes are prone to weight loss, decay, and softening when stored at an unsuitable temperature (*Sudheeran et al., 2018*). Therefore, storing mangoes at low temperatures is a commonly used technique to extend their shelf life. However, mangoes are highly perishable once kept at room temperature; thus, low-temperature storage without additional treatments is insufficient. So far, many alternative postharvest storage methods have been reported for extending the storage life of mango fruit after harvest, including the treatment with heat (*Pu et al., 2020*), cold shock (*Guo et al., 2020*), salicylic acid (SA) (*Vithana et al., 2019*), and nitric oxide (NO) (*Huang et al., 2020*), sodium alginate (*Rastegar, Hassanzadeh Khankahdani & Rahimzadeh, 2019*), and edible coating (*Sousa et al., 2021*).

Fruit can undergo oxidative stress during postharvest ripening and senescence, generating more reactive oxygen species (ROS). ROS are the by-products of metabolism, growth, and development and are distributed in various organelles (*Lu et al., 2009*; *Sachdev et al., 2021*). Plants have potential nonenzymatic and enzymatic mechanisms to maintain the balance between ROS generation and scavenging under physiological environments at a steady state (*Zhang et al., 2020*). However, certain abiotic stressors generated during oxidative stress can lead to excessive ROS production, causing indirect cell structure destruction, membrane lipid peroxidation, protein and DNA damage, and abnormal fruit ripening (*Gill & Tuteja, 2010*). Early responses in postharvest fruit are mainly caused by abnormalities in the system responsible for producing and scavenging ROS (*Zhao et al., 2006*; *Wantat, Seraypheap & Rojsitthisak, 2022*). Glutathione (GSH), a potent antioxidant, is an essential component of the ascorbate-glutathione (AsA-GSH) pathway of plant cells that scavenges ROS and protects the plant against oxidative damage (*Hasanuzzaman, Hossain & Fujita, 2011*; *Nahar et al., 2015*). The application of GSH to strawberry fruit could increase its endogenous antioxidant content and antioxidant capacity (*Ge et al., 2019*). In tomato seedlings, GSH treatment could induce antioxidant generation and activate the AsA-GSH cycle to alleviate oxidative damage and improve the salt stress response (*Zhou et al., 2017*). The application of cadmium (Cd) stress to plants could significantly inhibit their GSH biosynthesis, resulting in a significant increase in the levels of hydrogen peroxide ($H_2O_2$) and superoxide anion($O_2^-$) (*Xu et al., 2016*). A previous study has proved that GSH treatment could protect young loquat fruit against chilling stress (*Wu et al., 2011*).

AsA and GSH are nonenzymatic antioxidants, whereas superoxide dismutase (SOD), catalase (CAT), peroxidase (POD), and ascorbate peroxidase (APX) are antioxidant enzymes that can scavenge ROS. Additionally, the AsA-GSH cycle contributes to ROS scavenging during postharvest storage (*Song et al., 2016*). The AsA-GSH cycle includes a few reactions. APX first converts the intracellular $H_2O_2$ into water, where monodehydroascorbate (MDHA) produced from AsA serves as an electron donor. MDHA regenerates AsA through the action of monodehydroascorbate reductase (MDHAR), which spontaneously transforms into dehydroascorbate (DHA) (*Bartoli et al., 2017*). GSH subsequently reduces DHA to AsA, which is oxidized to glutathione disulfide (GSSG) by dehydroascorbate reductase (DHAR). Glutathione reductase (GR) then reduces GSSG to

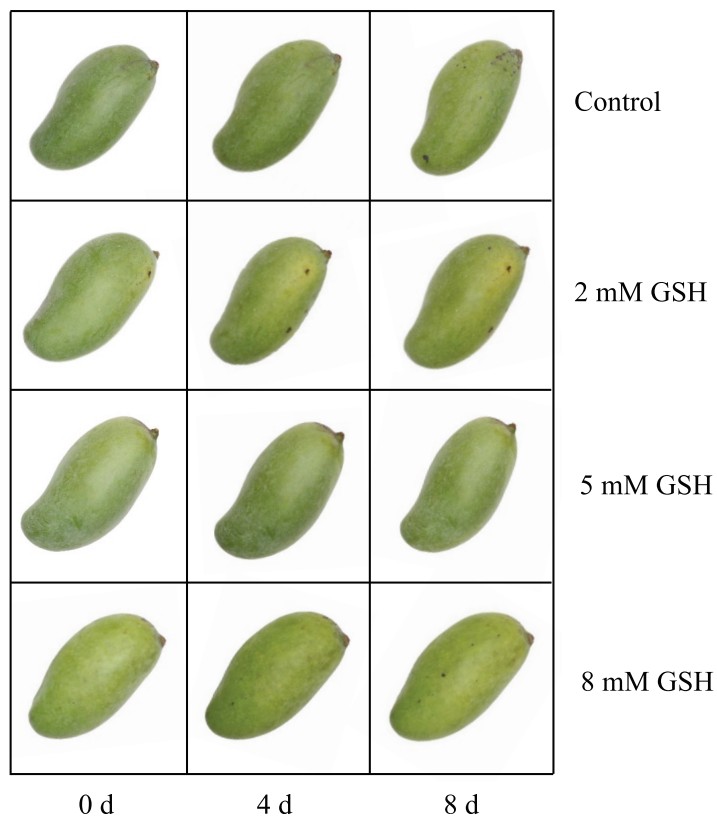

Control

2 mM GSH

5 mM GSH

8 mM GSH

0 d          4 d          8 d

**Figure 1 The phenotype in mango fruit under exogenous GSH during storage.** Disease spot color is black in mango fruit.                              

GSH. Plants with adequate levels of GSH and AsA, and APX, GR, MDHAR, and DHAR enzymes involved in the AsA-GSH pathway could tolerate oxidative stress (*Hasanuzzaman et al., 2019*). Many studies have shown that the AsA-GSH cycle plays a significant role in maintaining the postharvest quality of fruit and delaying their senescence (*Haroldsen et al., 2011*; *Min et al., 2020*; *Yan et al., 2022*).

In this study, the quality and antioxidant capacity of postharvest mango fruit treated with GSH was evaluated by measuring its physical and chemical quality, redox status, enzymatic activity, and gene expression levels. Additionally, the impacts of GSH treatment on the AsA-GSH cycle were explored by determining the activities of the key enzymes involved in the cycle and their gene expression levels. This study presents the empirical and theoretical foundation that can be beneficial for the development of preventative techniques against senescence in postharvest fruits.

## MATERIALS AND METHODS

### Materials and treatments

Mature green 'Guixiang' mangoes (*Mangifera indica* L.) (Fig. 1) were collected from a commercial farm in Zhanjiang, Guangdong Province, Southern China (110.34592E, 21.268491N). Six mangoes were packed in a bubble box in two layers to reduce the water loss and stacking damage, and newspapers were used as separators. The fruit was

transported 8.6 km from the harvest location to Lingnan Normal University. The selected process included 800 mangoes with uniform sizes and maturity and free of mechanical injury and diseases. A solution of 5% sodium hypochlorite and 0.01% chlorinated water was used to wash each mango fruit. The fruit was air-dried and subjected to various treatments, and each treatment had 200 mangoes: (1) Control: distilled water; (2) 2 mM GSH: 2 mmol/L GSH; (3) 5 mM GSH: 5 mmol/L GSH; and (4) 8 mM GSH: 8 mmol/L GSH. Tween-80 (1:1,000, v/v) was added to all solutions. After soaking for 5 min, the mango was naturally dried and stored at $25 \pm 1$ °C and 80–85% relative humidity (RH). The mango flesh was flash-frozen in liquid nitrogen on days 0, 4, 8, and 12 and stored at $-80$ °C until further analysis. On day 12, the fruit contained more than 80% of its original weight and was mushy, shriveled, and overgrown with fungal mycelium, so it was discarded from the experiments, as it became unsuitable to be sold. The experimental design was completely randomized using four biological replicates.

## Determination of weight loss, firmness, and color

The physiological weight loss (%) was determined following our previously reported method. The fruit firmness was expressed in newtons (N) based on a previous study (*Zhou et al., 2022a*). The color of the mangoes was evaluated using a colorimeter (Chroma meter CR400; Konica Minolta, Tokyo, Japan), and the values were recorded as $L^*$ (lightness), $a^*$ (greenness), $b^*$ (yellowness), and $h°$ (hue angle) (*Palou et al., 1999*).

## Determination of pH, total soluble solids (TSS), titratable acidity (TA), and soluble sugars

The pH and TSS content (%) were determined using a digital pH meter and a digital refractometer, respectively. The TA (%) and soluble sugar content were estimated following the previously reported method (*Horwitz, 1960*; *Zhou et al., 2022a*).

## Determination of lipid peroxidation (malondialdehyde [MDA]) and hydrogen peroxide ($H_2O_2$)

The MDA content was determined using the 2-thiobarbituric acid reactive substances method. The $H_2O_2$ content was determined according to the method described by *Zhou et al. (2017)*.

## Determination of redox state

The total ascorbate content (AsA plus DHA) was determined according to the method reported by *Costa, Gallego & Tomaro (2002)* with slight modifications. The DHA content was calculated by subtracting the AsA content from the total ascorbate content. The GSH content was determined as the sum of GSH and GSSG according to a previously reported method (*Yu et al., 2002*). After GSH was removed by 2-vinyl pyridine derivatization, the GSSG content was determined.

## Determination of SOD, POD, and CAT activity

SOD (EC 1.15.1.1) and POD (EC1.11.1.7) activities were determined following the previously reported method (*El-Shabrawi et al., 2010*; *Zhou et al., 2022a*). The CAT (EC

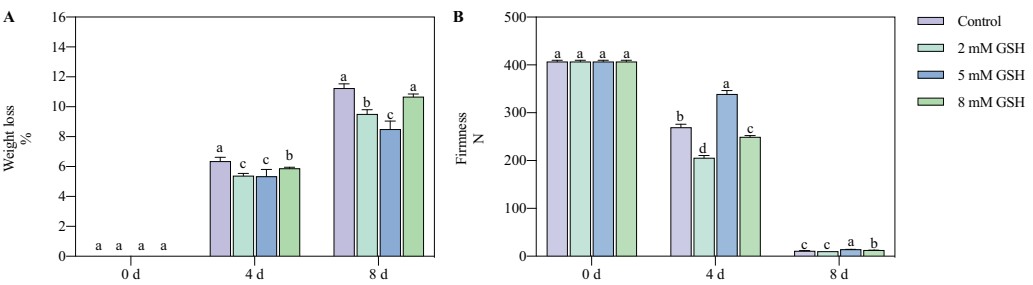

**Figure 2 Changes in weight loss (A) and firmness (B) in mango fruit under exogenous GSH during storage.** The error bars represent the standard error ($n = 4$). Bars with a different letter within a sampling date are significantly different ($P < 0.05$).

1.11.1.6) activity was measured according to a previously described method (*Hasanuzzaman, Hossain & Fujita, 2011*).

## Determination of APX, MDHAR, DHAR, and GR activity

The APX (EC: 1.11.1.11) (*Zhou et al., 2017*), MDHAR (EC: 1.6.5.4) (*Hossain, Nakano & Asada, 1984*), DHAR (EC: 1.8.5.1) (*Nakano & Asada, 1981*), and GR (EC: 1.6.4.2) (*Cakmak, Strbac & Marschner, 1993*) activities were measured according to previously reported methods.

## qRT-PCR analysis of gene expression levels

qRT-PCR analysis was performed according to our previously reported method (*Zhou et al., 2017*) using the primers listed in Table S1. The relative expression levels of *SOD*, *POD*, *CAT*, *APX*, *MDHAR*, *DHAR*, and *GR* genes in various samples were estimated using the $2^{-\Delta\Delta Ct}$ method (*Livak & Schmittgen, 2001*), and the reference gene was actin.

## Statistical analysis

The one-way analyses of variance (ANOVA) of the data were performed using SPSS 19.0 (SPSS Inc., Chicago, IL, USA) software. The statistical significance at $P < 0.05$ was determined using Tukey's test. A fully randomized design using four replicated samples was adopted.

# RESULTS

## Physical quality

### Weight loss

The postharvest weight loss of the mango fruit increased significantly with increased storage time (Fig. 2A). Compared with the control, the weight loss of the 2 mM GSH-treated mango fruit significantly decreased by 15.2% and 15.4% on days 4 and 8 of storage, respectively. The weight loss of the 5 mM GSH-treated mango significantly decreased at the storage time. Compared with the control, the weight loss of the 5 mM GSH-treated samples significantly decreased by 44.0% and 24.4% on days 4 and 8 of storage, respectively, whereas the weight loss of the 8 mM GSH-treated samples significantly decreased by 7.5% on day 4 of storage.

**Table 1 Changes in $L^*$, $a^*$, $b^*$, and $h°$ value in mango fruit under exogenous GSH during storage.**

| Glutathione concentration (mM) | Storage time (days) | | |
|---|---|---|---|
| | **0** | **4** | **8** |
| | *$L^*$ value* | | |
| 0 | 44.01 ± 0.50a | 51.78 ± 1.20a | 51.54 ± 4.31a |
| 2 | 44.01 ± 0.50a | 47.22 ± 1.42b | 50.70 ± 1.02ab |
| 5 | 44.01 ± 0.50a | 46.70 ± 1.14b | 44.89 ± 0.95c |
| 8 | 44.01 ± 0.50a | 48.30 ± 0.84b | 46.14 ± 0.64bc |
| | *$a^*$ value* | | |
| 0 | −16.44 ± 0.67a | −14.43 ± 1.58a | −14.00 ± 0.24a |
| 2 | −16.44 ± 0.67a | −16.06 ± 1.61a | −14.35 ± 0.28a |
| 5 | −16.44 ± 0.67a | −15.19 ± 1.03a | −14.19 ± 0.55a |
| 8 | −16.44 ± 0.67a | −16.26 ± 0.63a | −15.00 ± 0.52a |
| | *$b^*$ value* | | |
| 0 | 25.80 ± 1.22a | 32.67 ± 1.69a | 33.78 ± 0.83a |
| 2 | 25.80 ± 1.22a | 25.98 ± 2.39c | 31.72 ± 1.13a |
| 5 | 25.80 ± 1.22a | 27.76 ± 1.54bc | 28.52 ± 2.18b |
| 8 | 25.80 ± 1.22a | 30.49 ± 2.59ab | 28.55 ± 0.36b |
| | *$h°$* | | |
| 0 | 119.57 ± 0.25a | 116.31 ± 0.51b | 112.99 ± 1.54c |
| 2 | 119.57 ± 0.25a | 120.52 ± 0.82a | 115.45 ± 0.32b |
| 5 | 119.57 ± 0.25a | 116.37 ± 1.55b | 116.02 ± 0.29ab |
| 8 | 119.57 ± 0.25a | 117.53 ± 0.40b | 117.63 ± 1.16a |

**Note:**
Each value is the mean ± standard error ($n = 4$), and the error bars represent the standard error. Bars with a different letter within a sampling date are significantly different ($P < 0.05$).

### Firmness

The postharvest firmness of the mango fruit decreased with increased storage time (Fig. 2B). Compared with the control, the firmness of the 2 mM GSH-treated mangoes significantly decreased by 23.6% on day 4, whereas the firmness of the 8 mM GSH-treated significantly decreased by 7.4% on day 4 but increased by 17.0% on day 8. The treatment with 5 mM GSH could maintain the firmness of mangoes during all storage periods. The firmness of the 5 mM GSH-treated fruit increased by 25.9% and 30.7% on days 4 and 8 of storage, respectively.

### Peel color

Color is one of the major visual attributes of fruits. The peel color of the mango fruit used in this study was green when ripened. The control fruit showed a faster color change compared with the fruit treated with higher GSH concentrations (Table 1). The change in the peel color of fruit treated with higher GSH concentrations was slower, as indicated by the slower increase in its $L^*$ and $b^*$ values. On storage day 8, the $h°$ value of the peel of 5 mM GSH-treated fruit was higher compared with the control. The fruit samples were

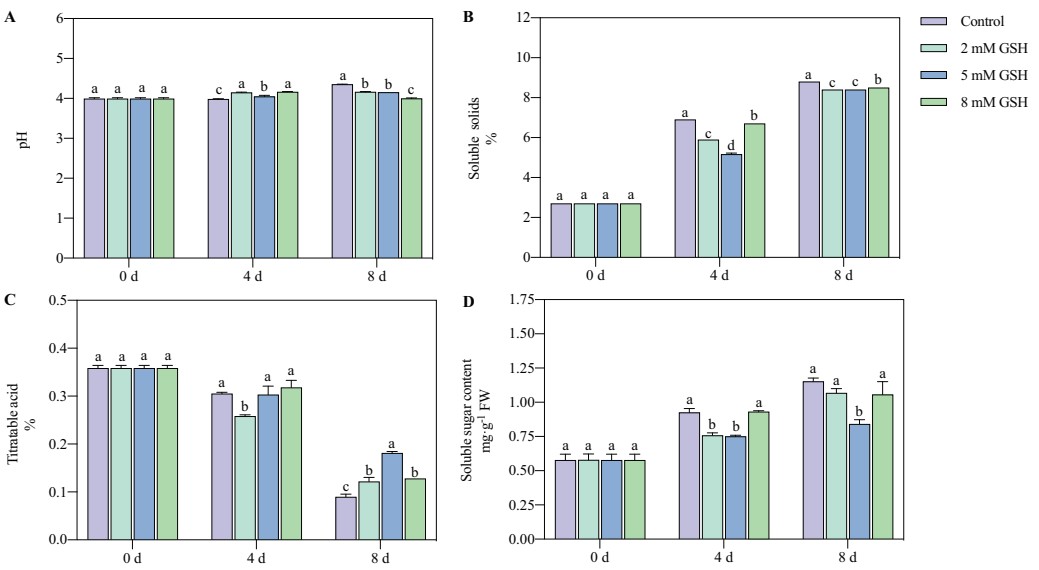

**Figure 3 Changes in pH (A), total soluble solids (TSS) (B), titratable acidity (TA) (C), and soluble sugar (D) in mango fruit under exogenous GSH during storage.** The error bars represent the standard error ($n = 4$). Values with a different letter within a sampling date are significantly different ($P < 0.05$).

olive-green, and no disease spots were observed on the peel of the 5 mM GSH-treated fruit at the end of storage (Fig. 1).

## Biochemical quality

The pH value of the non-treated mango fruit increased with increased storage time (Fig. 3A). Compared with the control, the pH value of the mango fruit treated with 2 mM GSH significantly increased by 4.1% on day four but decreased by 4.4% on day 8. The change in the pH of the mango fruit treated with 5 mM GSH was lower during the storage period. Compared with the control, the pH value of the 5 mM GSH-treated mango fruit was higher on day 4 and lower on day 8 by 0.04 fold. Compared with the control, the pH of the 8 mM GSH-treated fruit was unstable; it significantly increased by 4.4% on day 4 and decreased by 8.2% on day 8.

The postharvest TSS content in the mango fruit significantly increased during the storage period (Fig. 3B). The GSH-treated fruits better maintained their TSS content than the untreated fruit. Compared with the control, the TSS content in the mango fruit treated with 2 mM GSH decreased by 14.5% and 4.5% on days 4 and 8, respectively. Compared with the control, the TSS content in the 5 mM GSH treated fruit decreased by 25.1% and 4.5% on days 4 and 8 of storage, respectively. The postharvest TA content of mango fruit in all treatment groups gradually decreased during storage (Fig. 3C). Compared with the control, the TA content in the 2 mM of GSH-treated fruit significantly decreased by 15.3% on day 4 of storage but increased by 4.3% on day 8 of storage. Compared with the control, the TA contents in the 5 and 8 mM GSH treated fruits significantly increased by 115.1% and 50.0% on day 8 of storage, respectively. The soluble sugar content is a basic indicator of
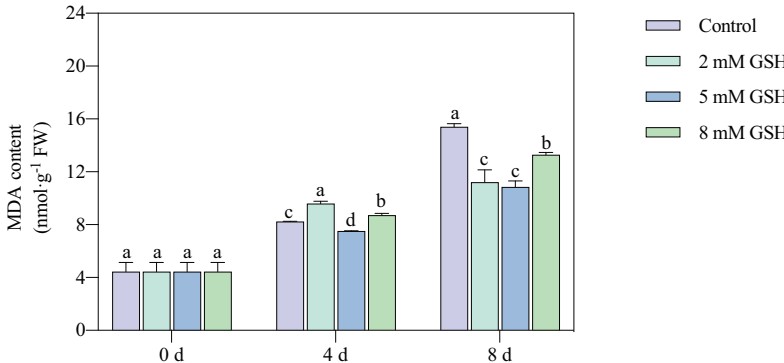

**Figure 4 Changes in MDA content in mango fruit under exogenous GSH during storage.** The error bars represent the standard error ($n = 4$). Bars with a different letter within a sampling date are significantly different ($P < 0.05$).

fruit ripening and stress resistance. The soluble sugar contents in all treatment groups increased during storage (Fig. 3D). Compared with the control, the soluble sugar content in the 2 mM GSH-treated fruit significantly decreased by 18.2% on day 4 of storage. Compared with the control, the soluble sugar contents in the 5 mM GSH treated fruit significantly decreased by 19.0% and 27.0% on days 4 and 8 of storage, respectively.

## Measurement of MDA content
Lipid peroxidation of the mango fruit in all treatment groups gradually increased with increased storage time (Fig. 4). The lipid peroxidation of fruit treated with 2 mM GSH was 0.2-fold higher on day 4 of storage and 0.3-fold lower on day 8 of storage compared with the control group. The 8 mM GSH treated fruit showed the same trend as the fruit treated with 2 mM GSH. However, compared with the control, the MDA content in the 5 mM GSH-treated fruit decreased by 8.7% and 29.5% on days 4 and 8 of storage, respectively. The 5 mM GSH treated fruit showed a decreased MDA content during storage.

## Measurement of H$_2$O$_2$ content
The H$_2$O$_2$ content in the 2 mM GSH-treated fruit and untreated fruit showed a similar trend; the content in both sample groups substantially increased during storage (Fig. 5). Compared with the control, the H$_2$O$_2$ content in the 5 mM GSH-treated fruit decreased by 46.1% and 33.3% on days 4 and 8 of storage, respectively. On the contrary, the H$_2$O$_2$ content in the 8 mM GSH treated fruit decreased by 24.3% during the mid-storage period compared with the control.

## Measurement of AsA and GSH contents, and AsA/DHA and GSH/GSSG ratios
In all treatment groups and during all storage periods, the AsA content first increased and then decreased, while the GSH content steadily increased (Fig. 6). The AsA/DHA ratio of the 5 mM GSH-treated fruit increased with increased storage time, while it decreased in other groups. The GSH/GSSG ratio of the 5 mM GSH-treated fruit gradually increased during storage. Compared with the control, the AsA content of the 2 mM GSH-treated

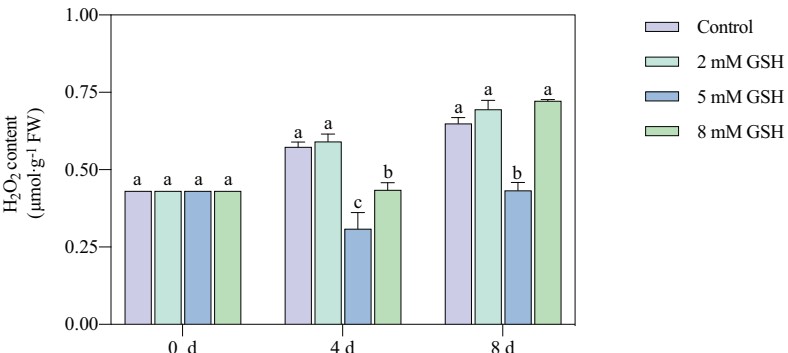

**Figure 5 Changes in H₂O₂ content in mango fruit under exogenous GSH during storage.** The error bars represent the standard error (*n* = 4). Bars with a different letter within a sampling date are significantly different (*P* < 0.05).

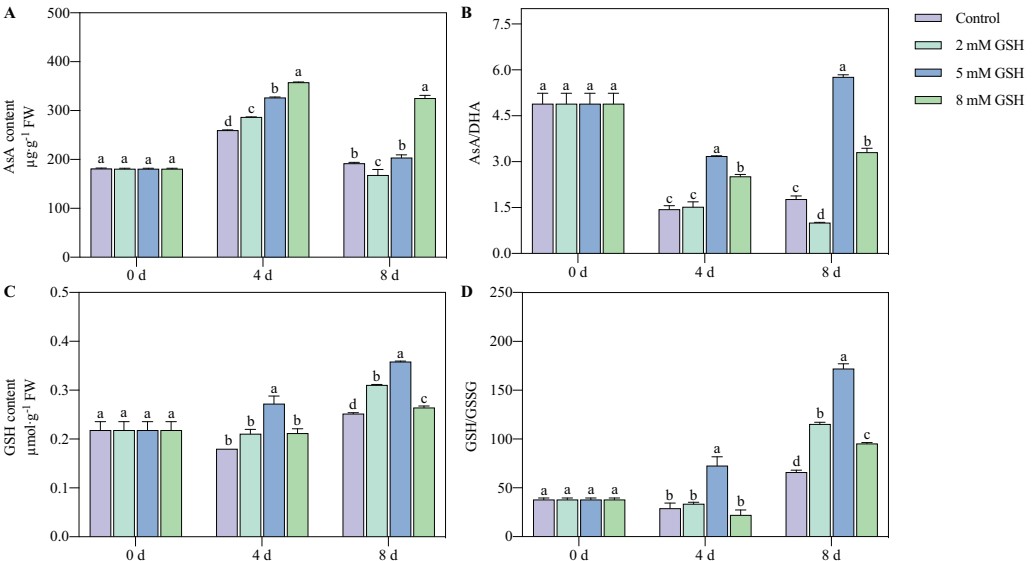

**Figure 6 Changes in redox status in mango fruit under exogenous GSH during storage.** The error bars represent the standard error (*n* = 4). Bars with a different letter within a sampling date are significantly different (*P* < 0.05).

fruit significantly increased by 10.6% on day 4, the AsA content and AsA/DHA ratio decreased by 12.2% and 43.5%, respectively, and the GSH content and GSH/GSSG ratio increased by 23.3% and 74.5%, respectively, on day 8. Compared with the control, the AsA content in the 5 mM GSH treated fruit significantly increased by 26.1% on day 4, and the GSH content, AsA/DHA ratio, and GSH/GSSG ratio increased by 51.2%, 120.9%, and 149.5%, respectively, on day 4, and 42.4%, 224.5%, and 160.3%, respectively, on day 8. Compared with the control, the AsA content and AsA/DHA ratio in the 8 mM GSH treated fruit significantly increased by 38.1% and 74.9%, respectively, on day 4 of storage, and the GSH content and GSH/GSSG ratio increased by 4.9% and 44.3%, respectively, on day 8 of storage.

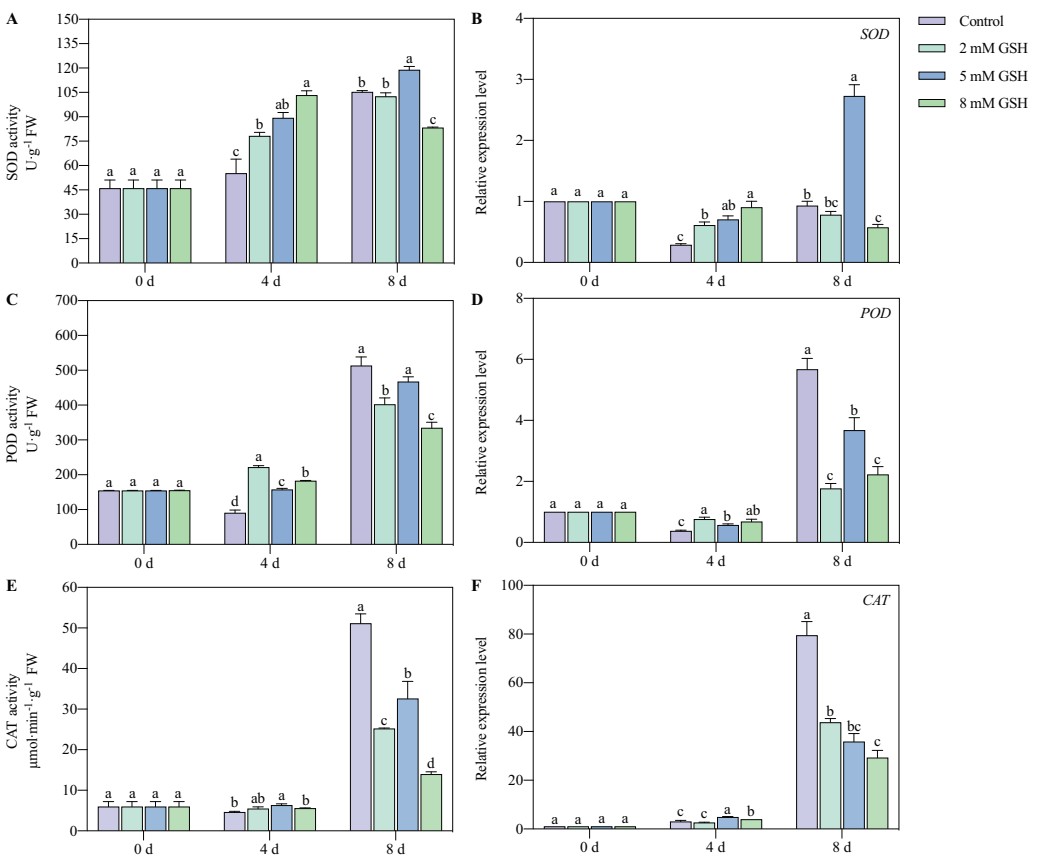

**Figure 7 Changes in activities and gene expression levels of SOD (A and B), POD (C and D), CAT (E and F) in mango fruit under exogenous GSH during storage.** The error bars represent the standard error ($n = 4$). Bars with a different letter within a sampling date are significantly different ($P < 0.05$).

## Activities of SOD, POD, and CAT and their gene expression levels

The SOD and POD activities of the 2 and 8 mM GSH-treated mango fruit were higher than those in control on day 4 of storage (Figs. 7A, 7C, and 7E). The SOD activity of the 8 mM GSH-treated fruit and the POD and CAT activities of the 2 and 8 mM GSH-treated fruit were significantly lower than those in control on day 8 of storage. The SOD activity of the 5 mM GSH treated fruit significantly increased by 66.2% and 12.9% on days 4 and 8 of storage, respectively. The POD and CAT activities increased by 74.0% and 37.1%, respectively, on day 4 of storage, and the CAT activity decreased by 36.3% on day 8 of storage, compared with the non-treated fruit. On day 4 of storage, the expression levels of the *SOD*, *POD*, and *CAT* genes in all groups of the GSH-treated mango fruit (except for that of *CAT* in 2 mM GSH-treated fruit) significantly up-regulated compared with the control group (Figs. 7B, 7D, and 7F). In contrast, the expression levels of the *SOD*, *POD*, and *CAT* genes in the GSH-treated fruit (except for that of *SOD* in 5 mM GSH-treated fruit) were lower than those in control on day 8 of storage.

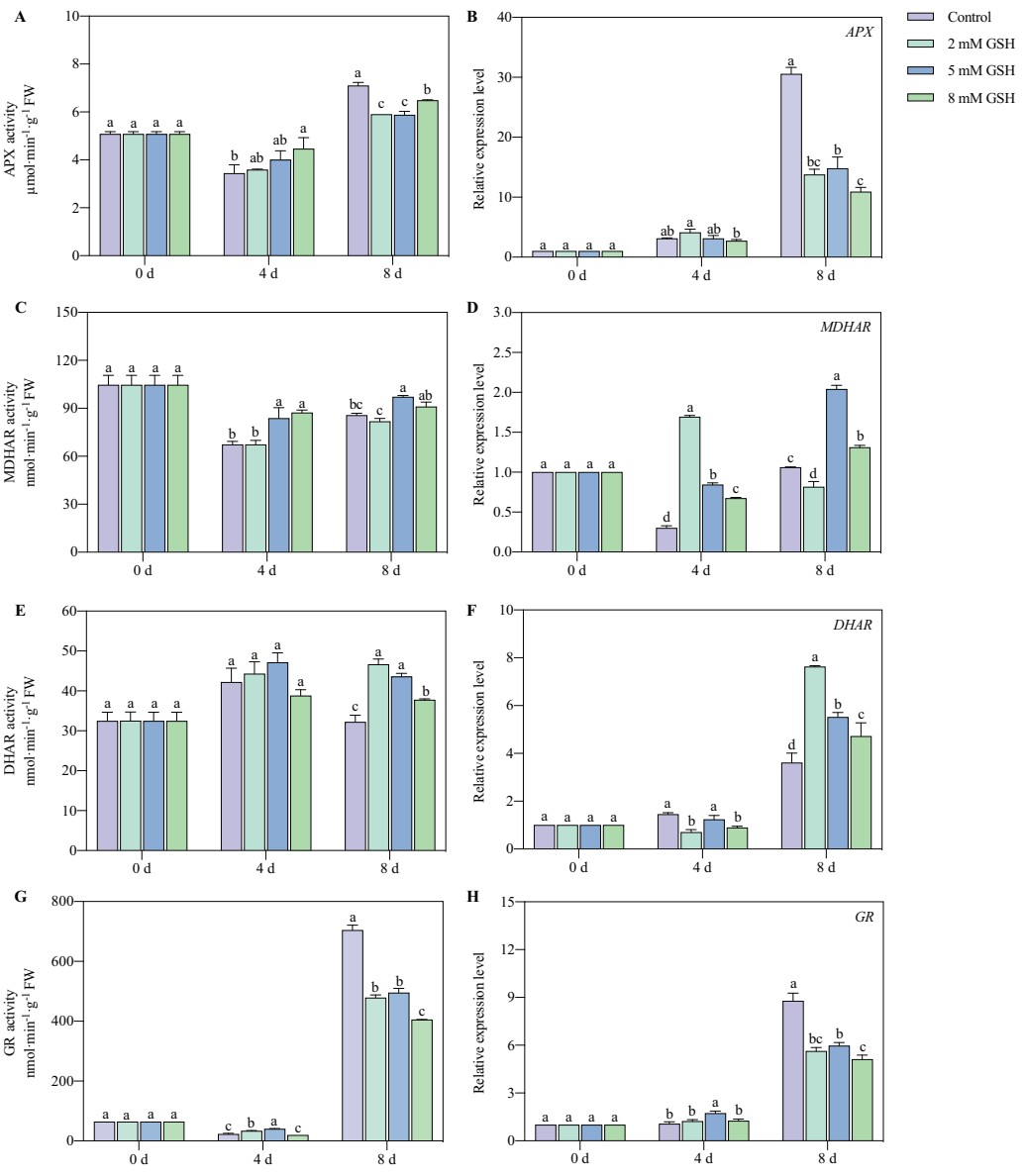

**Figure 8 Changes in activities and gene expression levels of APX (A and B), MDHAR (C and D), DHAR (E and F) and GR (G and H) in mango fruit under exogenous GSH during storage.** The error bars represent the standard error ($n = 4$). Bars with a different letter within a sampling date are significantly different ($P < 0.05$).

## Activities of key enzymes in the AsA-GSH cycle and their gene expression levels

The APX and MDHAR activities of the 8 mM GSH-treated fruit and the GR activity of the 2 and 5 mM GSH-treated fruit were significantly higher compared with the control group on day 4 of storage (Figs. 8A, 8C, and 8G). Compared with the control, the APX and GR activities were significantly lower on day 8 of storage, whereas the DHAR activity was significantly higher in the GSH-treated fruit (Figs. 8A, 8E and 8G). The MDHAR activity of the 5 mM GSH significantly increased by 24.5% and 13.3% on days 4 and 8 of storage,

respectively. Similarly, the DHAR activity increased by 35.3% on day 8, and the GR activity increased by 78.0% on day 4, but the APX and GR activities decreased by 17.2% and 29.7% on day 8 of storage.

The expression level of *MDHAR* and *DHAR* in the GSH treated fruits at different concentrations significantly increased on days 4 and 8, but the expression level of *APX* and *GR* decreased on day 8, respectively, compared with the control (Figs. 8B, 8D, 8F and 8H). The fruit treated with 2 mM GSH had a lower expression of *MDHAR* and *DHAR* on days 8 and 4 of storage compared with the control. The expression level of *MDHAR* increased on day 8, and the expression level of *DHAR* decreased on day 4 in the 8 mM GSH-treated fruit compared with the control. The fruit treated with 5 mM GSH had the highest *MDHAR* transcript levels among all treated fruit in other groups on day 8, and the expression of *GR* in this group of fruit increased by 0.62 fold compared with the control on day 4 of storage.

## DISCUSSION

GSH provides a barrier by creating a semipermeable layer on the fruit surface to reduce transpiration losses. In tomatoes (*Zhou et al., 2022b*) and strawberries (*Ge et al., 2019*), GSH acts as a barrier against water loss to reduce weight loss. The present study results confirmed that the treatment with 5 mM GSH could significantly prevent weight loss in mangoes. However, the treatment with higher GSH (8 mM) concentrations at the end of storage time was not significant for weight loss compared with the control, and the reason needs further exploration. During storage, the cell wall components of the peel disintegrate, the water content changes, and the hardness decreases, causing the fruit to lose its texture, thereby losing its market value (*Tharanathan, Yashoda & Prabha, 2006*). GSH acts as an antioxidant that can retain its antioxidant ability after forming an edible coating on the fruit peel. For instance, AsA, an antioxidant, can maintain the postharvest firmness of fruit (*Sogvar, Koushesh Saba & Emamifar, 2016*). In a previous study, 100 mM of GSH decreased the firmness of strawberry fruit, which was not significantly different from the control (*Ge et al., 2019*). This might be due to low metabolic activities and reduced respiration rates caused by GSH coating; thus, the pH remained unchanged. Fruits treated with edible coating have a higher level of citric acid, with lower respiration rates, fungal infestation, and longer shelf life (*Han et al., 2004*; *Saleem et al., 2021*). Mango fruit treated with 5 mM of GSH maintained low TSS content on day 4 of storage. Although the TSS content in the 8 mM GSH-treated fruit decreased compared with the control, the decrease was not as drastic as that in other groups of GSH-treated fruit. In a previous study, the GSH-treated fruit, such as the AsA-treated fruit, was protected by a semi-permeable film, which reduced $O_2$ and/or elevated $CO_2$, suppressed ethylene evolution, and altered the internal atmosphere (*Singh & Mirza, 2018*). Additionally, the fruit with a greater weight loss has a higher TSS content due to their edible coating, such as chitosan, which can act as a barrier to prevent water loss and maintain the TSS content in the fruit (*Ackah et al., 2022*). The 5 mM GSH coating showed the same function in mangoes. The fruit treated with 5 mM GSH had the highest TA content at the end of storage. The overall decline in organic acids can be predicted by metabolic changes in the fruit or by the consumption of organic acids during respiration (*Yuan, Chen & Li, 2016*;

*Gomes et al., 2020*). Thus, the 5 mM GSH coating probably reduced the TA loss by decreasing the metabolic activities. Compared with the uncoated mangoes, the reduction in soluble sugar contents in mangoes coated with hydroxypropyl methylcellulose was lower (*Sousa et al., 2021*). A sufficient GSH concentration could also inhibit the rapid increase in sugar content during fruit ripening.

A semipermeable edible coating can limit the oxygen supply to fruit tissues, reducing lipid peroxidation and alleviating oxidative damage during postharvest storage (*Xu et al., 2012*; *Li et al., 2019*). Antioxidants, such as AsA, can palliate oxidative stress in the tissues of fresh produce (*Ali et al., 2020*). At appropriate concentrations, the membrane integrity of the coated fruit can be maintained in the GSH treated group, thereby decreasing the MDA levels. As one of the well-known ROS responsible for oxidative damage in fresh produce, $H_2O_2$ is formed due to postharvest storage conditions (*Imahori, Bai & Baldwin, 2016*; *Luo et al., 2020*; *Palma et al., 2020*). The produced ROS stimulates the antioxidant system, which helps to reduce the oxidative damage (*Sachdev et al., 2021*; *Pan et al., 2022*). During cold storage, GSH is involved in the AsA-GSH cycle, playing an important role in scavenging ROS (*Yao et al., 2021*). Since GSH can inhibit the production of $H_2O_2$, treating mango with GSH can alleviate oxidative damage during storage.

The POD, SOD, and CAT enzymes play an important role in preventing or alleviating the damage caused by ROS (*Rastegar, Hassanzadeh Khankahdani & Rahimzadeh, 2019*). Our previous study found that the treatment with 5 mM GSH could increase the SOD, POD, and CAT activities and the expression of *POD51/52/63* to eliminate ROS during postharvest tomato fruit ripening (*Zhou et al., 2022b*). According to *Yao et al. (2021)*, the edible GSH coating could preserve the antioxidant enzyme activity in bell pepper fruit. The increased expression levels of *SOD* and *CAT3* in the endosperm of cherries could lead to their best physiological quality (*Santos et al., 2014*). In this study, the treatment with 5 mM GSH increased the SOD activities during the entire storage period and increased the POD and CAT activities during the mid-storage period, and the expressions of genes encoding these enzymes showed a similar trend.

A decrease in the AsA content during storage could be caused by the conversion of dehydroascorbic acid or AsA oxidation (*Singh & Mirza, 2018*). The intracellular redox state plays an important role in maintaining the biotic and abiotic stresses (*Udenigwe et al., 2016*). AsA has been shown to enhance vitamin C retention in ber fruit (*Singh & Mirza, 2018*). The fruit needs to maintain its high reducing power to quench excess ROS (*Peng et al., 2022*). Since GSH is an important cellular antioxidant, GSH concentration and the GSH/GSSG ratio act as determinants of the cellular redox state and indirectly affect several essential cellular activities (*López-Huertas & Palma, 2020*; *Ding et al., 2021*). In postharvest fruit, the high GSH content and GSH/GSSG ratio could maintain the balance of the redox environment and reduce oxidative stress during fruit ripening and senescence (*Mellidou et al., 2012*; *Lin et al., 2020*). It has been shown that exogenous GSH could improve the cell membrane properties (*Nemat Alla & Hassan, 2014*), and an increase in endogenous GSH levels could reduce oxidative stress caused by abiotic stresses. In this study, the application of an appropriate amount of GSH induced the antioxidant metabolite accumulation, providing resistance to storage-induced ROS generation in fruit.

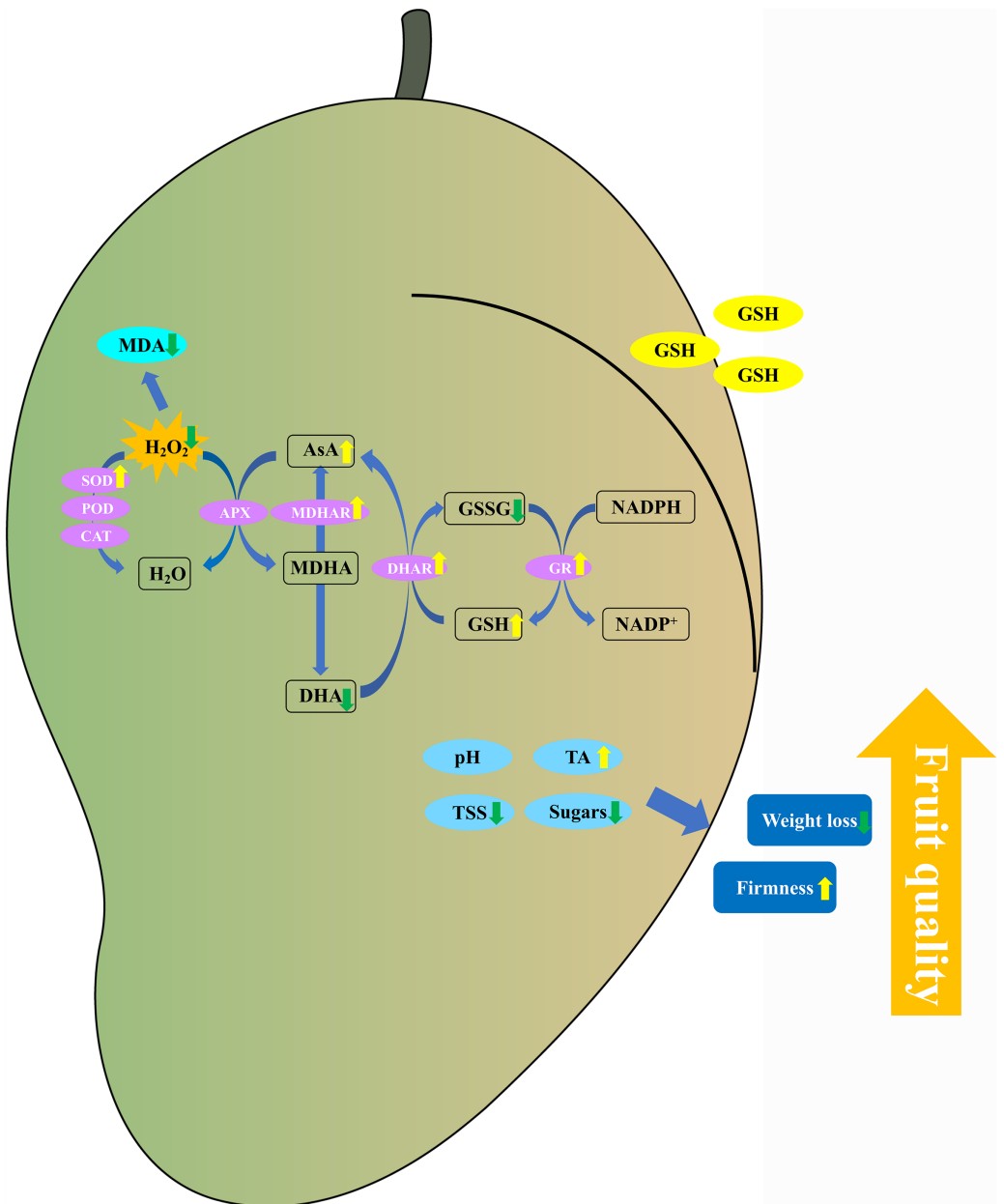

**Figure 9** Putative mechanism of exogenous GSH on antioxidant capacity in postharvest mango fruit.

Under an extended storage period, the APX, GR, MDHAR, and DHAR activities of the control fruit tend to decrease (*Zhao et al., 2009*). As a result, ROS formation and oxidative stress were exacerbated due to the decrease in antioxidant content and enzyme activity. In bell pepper fruit, the levels of APX, GR, and MDHAR activities were enhanced with exogenous GSH treatment, which could increase the endogenous GSH levels (*Yao et al., 2021*). Thus, exogenous GSH could induce the ROS scavenging system by accelerating the AsA-GSH cycle (*Nemat Alla & Hassan, 2014*). Several studies have reported the protective effects of GSH on tomatoes (*López-Vidal et al., 2016*), mango (*Zhao et al., 2009*), bell

pepper (*Yao et al., 2021*), and apple (*Davey & Keulemans, 2004*). The physiological performance of fruit improved with the increase in GSH level.

In the plant, the changes in gene expression are an indication of oxidative damage under oxidative stress (*Kang et al., 2013*). Previous research has found that peaches treated with hypobaric oxygen could reduce cold stress by upregulating the expression of *APX* and *MDHAR* and activating the AsA-GSH cycle (*Song et al., 2016*). In this study, *MDHAR* was up-regulated on day 8 in the 8 mM GSH-treated fruit. Over-expressing *SlMDHAR* and *SlDHAR* could alter the AsA levels to regulate fruit ripening (*Haroldsen et al., 2011*). The highest *MDHAR* transcript levels were in the 5 mM GSH-treated fruit. Under stressful environmental conditions, GR is essential for maintaining the redox state and detoxifying ROS in horticulture crops (*Zhou et al., 2017*). GR overexpression can maintain cellular redox and decrease oxidative damage in wheat (*Melchiorre et al., 2009*). In the present study, the treatment with 5 mM GSH strongly upregulated the *GR* expression in postharvest mango fruit during the mid-storage period but substantially downregulated the *GR* expression with prolonged storage period. This might be associated with the decrease in the antioxidant content and enzyme activities involved in the AsA-GSH cycle. During storage, the presence of GSH could prolong the MDHAR and DHAR activities, suppress ROS, maintain the integrity of cell structures, and delay senescence in mango fruit. Based on the present study results, the upregulation of *MDHAR*, *DHAR*, and *GR* increased the reduced AsA and GSH contents and the activities of enzymes in the AsA-GSH cycle, thereby decreasing oxidative damage and lipid peroxidation. These changes were observed with improved protection of mango fruit against oxidative stress during postharvest storage due to the treatment with exogenous GSH. Exogenous GSH could induce the AsA-GSH cycle and protect postharvest mango fruit from ROS-caused ripening and senescence during storage.

## CONCLUSIONS

In conclusion, the 5 mM GSH treatment significantly maintained the postharvest quality of mango fruit by decreasing the weight loss, TSS, and soluble sugar content and increasing the firmness and TA content (Fig. 9). Additionally, the 5 mM GSH treatment significantly decreased the MDA and $H_2O_2$ by increasing the activities of SOD, MDHAR, DHAR, and GR enzymes and their gene expression levels, which accelerated the AsA-GSH cycle. This phenomenon increased the AsA and GSH levels, maintained the cellular redox status balance, and alleviated oxidative stress during storage. Overall, GSH could improve the antioxidant capacity and maintain the postharvest quality of mango fruit.

The experimental results provide insights into the development of new preservation technologies.

### Funding

This work was supported by the Open Project Fund of Hainan Key Laboratory of Storage & Processing of Fruits and Vegetables (No. HNGS202104 and No. HNGS202302), the

National Natural Science Foundation of China, China (No. 32202476), the Natural Science Foundation of Guangdong, China (No. 2022A1515010719), the Program for Key Areas of Universities in Guangdong Province (No. 2021ZDZX4035), the Science and Technology Special Fund Project of Guangdong Province (No. 2021A05192 and No. 2021A05222), the Lei Yang Academic Posts Programmer of Lingnan Normal University (No. 2022).
The funders had no role in study design, data collection and analysis, decision to publish, or preparation of the manuscript.

## Grant Disclosures

The following grant information was disclosed by the authors:
Open Project Fund of Hainan Key Laboratory of Storage & Processing of Fruits and Vegetables: HNGS202104 and HNGS202302.
National Natural Science Foundation of China: 32202476.
Natural Science Foundation of Guangdong, China: 2022A1515010719.
Program for Key Areas of Universities in Guangdong Province: 2021ZDZX4035.
Science and Technology Special Fund Project of Guangdong Province: 2021A05192 and 2021A05222.
Lei Yang Academic Posts Programmer of Lingnan Normal University: 2022.

## Competing Interests

The authors declare that they have no competing interests.

## Author Contributions

- Yan Zhou conceived and designed the experiments, prepared figures and/or tables, authored or reviewed drafts of the article, and approved the final draft.
- Jiameng Liu analyzed the data, authored or reviewed drafts of the article, and approved the final draft.
- Qiongyi Zhuo analyzed the data, prepared figures and/or tables, and approved the final draft.
- Keying Zhang analyzed the data, prepared figures and/or tables, and approved the final draft.
- Jielin Yan analyzed the data, prepared figures and/or tables, and approved the final draft.
- Bingmei Tang analyzed the data, prepared figures and/or tables, and approved the final draft.
- Xiaoyun Wei analyzed the data, prepared figures and/or tables, and approved the final draft.
- Lijing Lin performed the experiments, analyzed the data, authored or reviewed drafts of the article, and approved the final draft.
- Kaidong Liu performed the experiments, authored or reviewed drafts of the article, and approved the final draft.

## Data Availability

The raw measurements are available in the Supplemental Files.

## Supplemental Information

Supplemental information for this article can be found online at http://dx.doi.org/10.7717/peerj.15902#supplemental-information.

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
