# Peer review of "Exogenous glutathione maintains the postharvest quality of mango fruit by modulating the ascorbate-glutathione cycle"

_PeerJ, doi:10.7717/peerj.15902_

## Round 0.1 · original submission · Major Revisions

Authors are advised to carefully address all the points raised by reviewers. Also, improve the language of the manuscript.

Reviewer 1 ·

Basic reporting

The manuscript by Zhou et.al. has tested the exogenous treatment of GSH in extending post-harvest quality of mango fruits. The study provided useful knowledge for developing novel techniques to extend shelf life and reduce product losses.

However, there are a few issues in the draft. Hope the authors could address before considered for publication:
1. In each figure, day 0 had the exact same value for all four treatment groups. Does this mean that the authors quantified the day 0 metrics of all groups together? It would mask possible variations in each treatment group that may contribute to later differences. I would suggest the authors measure each group independently from day 0.
2. Please provide more details in the method section. For instance, line 123-126 was difficult to understand. Would be helpful with more description as how to interpret the L,a,b,h values.
3. The change in color was subtle in Figure 1. Could the authors mark or highlight regions that showed most differences, as well as disease spots?
4. Some of the metrics in Control group showed reversed levels in day 4 and day 8 (i.e., Fig. 6a, Fig. 7b-7d, Fig. 8), making it difficult to interpret the effect of GSH treatment. Could the authors provide any explanation? Or maybe adding additional time points to clearly show the trend of changes?
5. Could the author include over-ripe stage (Day 12 or later) to show more drastic effect of GSH in preserving product quality?

Experimental design

no comment

Validity of the findings

no comment

Reviewer 2 ·

Basic reporting

I appreciate the authors for your efforts to improve this interesting article. However, I think it is very opportune to make significant corrections, a major revision of the manuscript. “Exogenous glutathione maintains the postharvest quality of mango fruit by modulating the ascorbate-glutathione cycle”. This manuscript investigated maintains the postharvest quality of mango fruit by modulating the ascorbate-glutathione cycle from a commercial farm in Zhanjiang, Guangdong Province, Southern China and provided a case study of this kind of research.
As per the current form, the manuscript has several issues which need to improve greatly. I have read the entire manuscript and my initial comment is that manuscript is poorly written. I have significant concerns about the grammar and vocabulary of the manuscript; therefore, improvement of the language is highly needed.
The questions that should also be explained by the author is What are the implications for government policy? How can the results of this research benefit policy makers? how can this novelty of research provide input on the development of science, especially in the field of science?
How can policy makers use the results of this study for the current status and future invasion risk?

Experimental design

1) Abstract:
why to study this? The structure of the abstract should be improved, as well as the lack of several aspects that should be included in this section. Most of the abstract contain confusing and uninformative sentences. Please give more precise objectives here (such as in the Abstract) and put the main objective in the abstract. In Abstract section as subheading Background does not explain the year of study, research station, duration? Sub section as Methods., does not explaining the study design or model. Sub section as Conclusions, lack of future recommendations.
Keywords do not fully reflect the essence of the article; it is necessary to specify
2) Introduction:
Introduction is not properly contextualized. It needs a great improvement by explaining both merits and demerits of postharvest quality and modulating the ascorbate-glutathione cycle.
Introduction grammatical issues appear to be most prevalent in the introduction, making for very confusing reading. Further, the introduction is long but has no clear thread.
The overall structure of introduction section is a mess. Authors need to re-structure it and present in a systematic way. Start by global perspective of postharvest quality of mango fruit and why it is an important issue. Then move to what is happening in China and why it matters when it comes to Exogenous glutathione maintains and postharvest quality of mango fruit.
Then introduce literature in China regarding how modulating the ascorbate-glutathione cycle have been improving postharvest quality of mango fruit and what are the gaps.
Based on those gaps, introduce your objectives.
Close the introduction with objectives to answer the questions and what will be the benefits of answering them.
3) Materials and Methods:
The authors have to provide study area map and more details.
Also provide lab work pictorial as evidence which type and stages of working completed.
The authors have to provide methodology layout as flow sheet diagram.
The authors have to add more citation in Materials and Methods section

Validity of the findings

4) Results:
The discussion of the research results is insufficient and does not refer to other studies conducted in this area. This chapter is more like a summary than a discussion of the results. I am not agreed to results interpretation and discussion sections should be single format, please make two sections to modify. Make one hypothetical figure, which depicts the findings of this study.
5) Discussion:
Discussion needs improvement. Also, write limitations of the study.
6) Conclusions:
Conclusions should be brought closer to the results of research - to make them more specific. Conclusion is insufficient in present form, it should be improved by adding decision-makers role in this context, write a few sentences the benefits of this research for the government, society and scientific development.

Additional comments

7) General comments:
The layout of the article should be revised.
Data interpretation and presentation should be revised.
Finally, ask the authors for an exhaustive review of the list of references.
The English language needs major editing.
The manuscript could be accepted once these changes are made.

Annotated reviews are not available for download in order to protect the identity of reviewers who chose to remain anonymous.

---

## Round 0.2 · accepted · Accept

The authors have revised the manuscript as per the suggestions of the reviewers.

Reviewer 1 ·

Basic reporting

The authors have addressed the comments with relevant references and revised the manuscript accordingly. No further comments.

Experimental design

no comment

Validity of the findings

no comment.